# Association of ABO blood group with COVID-19 severity, acute phase reactants and mortality

Uzma Ishaq[1], Asmara Malik[2], Jahanzeb Malik[3,4]*, Asad Mehmood[3], Azhar Qureshi[5], Talha Laique[6], Syed Muhammad Jawad Zaidi[7], Muhammad Javaid[3], Abdul Sattar Rana[3,4]

1 Department of Hematology, Foundation University Medical College, Islamabad, Pakistan, 2 Department of Community Medicine, National University of Medical Sciences, Rawalpindi, Pakistan, 3 Department of Cardiology, Rawalpindi Institute of Cardiology, Rawalpindi, Pakistan, 4 Department of Cardiology, Advanced Diagnostics and Liver Center, Rawalpindi, Pakistan, 5 Department of Cardiology, Armed Forces Institute of Cardiology, Rawalpindi, Pakistan, 6 Department of Pharmacology, Allama Iqbal Medical College, Lahore, Pakistan, 7 Department of Medicine, Rawalpindi Medical University, Rawalpindi, Pakistan

* heartdoc86@gmail.com

**Data Availability Statement:** All relevant data are within the paper and its Supporting Information files.

## Abstract

## Introduction

Coronavirus disease 2019 (COVID-19) is the ongoing pandemic with multitude of manifestations and association of ABO blood group in South-East Asian population needs to be explored.

## Methods

It was a retrospective study of patients with COVID-19. Blood group A, B, O, and AB were identified in every participant, irrespective of their RH type and allotted groups 1, 2,3, and 4, respectively. Correlation between blood group and lab parameters was presented as histogram distributed among the four groups. Multivariate regression and logistic regression were used for inferential statistics.

## Results

The cohort included 1067 patients: 521 (48.8%) participants had blood group O as the prevalent blood type. Overall, 10.6% COVID-19-related mortality was observed at our center. Mortality was 13.9% in blood group A, 9.5% in group B, 10% in group C, and 10.2% in AB blood group (p = 0.412). IL-6 was elevated in blood group A (median [IQR]: 23.6 [17.5,43.8]), Procalcitonin in blood group B (median [IQR]: 0.54 [0.3,0.7]), D-dimers and CRP in group AB (median [IQR]: 21.5 [9,34]; 24 [9,49], respectively). Regarding severity of COVID-19 disease, no statistical difference was seen between the blood groups. Alteration of the acute phase reactants was not positively associated with any specific blood type.

**Funding:** The author(s) received no specific funding for this work.

**Competing interests:** The authors have declared that no competing interests exist.

## Conclusion

In conclusion, this investigation did not show significant association of blood groups with severity and of COVID-19 disease and COVID-19-associated mortality.

## Introduction

Coronavirus disease 2019 (COVID-19) represents a public health emergency causing economic and health care system collapse worldwide. As of April 2021, approximately four million people have died from COVID-19 with the numbers increasing exponentially [1]. With some of the regions surviving through the fourth wave of the pandemic, there has been a multitude of manifestations associated with COVID-19, including the cardiovascular, respiratory, gastrointestinal, and hematological systems [2–4].

Apart from documentary evidence of auto-antigenicity and reporting of hematological complications, many studies have investigated the pathway to viral entry into the human hosts [5, 6]. One such hypothesis resides in the ABO blood group and its association with the severity of diseases such as Hepatitis B Virus (HBV), Middle-Eastern Respiratory Syndrome Coronavirus (MERS), and Severe Acute Respiratory Syndrome Coronavirus (SARS) [7]. Furthermore, several investigations have investigated an association of ABO blood group with COVID-19 [8]. A study on COVID-19 demonstrated a cross-replicating association signal at locus 9q34.2, which coincides with the ABO blood group locus [9]. However, the association between the blood groups and the severity of the disease is still unclear. Several studies have shown an increased risk of infectivity with blood group A while conferring a low risk of COVID-19 infection with blood group O [10]. A couple of systematic reviews demonstrate blood group A individuals' vulnerability to COVID-19, and blood type AB conferring a lower risk of SARS-COV-2 infection [11, 12].

In South-East Asia however, blood group O and B is prevalent and studies demonstrating the severity of COVID-19 disease and COVID-19-associated mortality in this population subset were lacking [13]. Hence, we conducted a retrospective analysis on a cohort of COVID-19 patients, analyzing their ABO blood types, and observed overall COVID-19-associated mortality and severity of disease in association with the blood type.

## Methods

This was a retrospective investigation conducted at Advanced Diagnostics and Liver Center after approval from the ethical review board of our institute (ID: ADC/17/20) according to the Declaration of Helsinki. All participants or their guardians gave written informed consent before data collection. A total of 1067 COVID-19 patients were included in this study from April 2020 through January 2021 and the demographic data, comorbid conditions, epidemiological data, laboratory tests, hospital stay, and mortality rates were extracted via the electronic system of our institute. Every patient was confirmed positive SARS-COV-2 via real-time reverse transcription-polymerase chain reaction (RT-PCR). The severity of COVID-19 was classified into, mild, moderate, and critical according to the Centers for Disease Control (CDC) guidelines. Individuals who have various signs and symptoms of COVID-19 (e.g., fever, cough, sore throat, headache, malaise, nausea, vomiting, loss of taste and smell, etc.) but do not have shortness of breath, dyspnea, or abnormal chest imaging were mild and those showing evidence of lower respiratory illness during clinical assessment or imaging and who

have an oxygen saturation (SpO$_2$) $\geq$ 94% were labelled moderate disease. Patients with SpO$_{2 < 94\%}$ on room air, respiratory rate of > 30 breaths/min, or > 50% lung infiltrates, and those requiring mechanical ventilation were labeled as critical. Patients with known hemoglobinopathies or other blood disorders were excluded [14].

All the tests were carried out in the clinical laboratory of Advanced Diagnostics under the standard procedures according to Punjab Health Commission. Hemoglobin (Hgb), and white blood cells (WBC) were performed on an XN-3100 Sysmex hematology analyzer. C-reactive proteins and D-dimers were analyzed on Cobas® c3011 analyzer (Roche Diagnostics) and interleukin-6 (IL-6), and Procalcitonin was analyzed via electrochemiluminescent immunoassay (ECLIA) in the Elecsys® 2010 immunoassay system. ABO blood group and the cross-match were done manually by an expert hematologist (UI). Group A, B, O, and AB were identified in every participant, irrespective of their RH type and allotted groups 1, 2,3, and 4 respectively.

Statistical analysis was carried out with Statistical Package for the Social Sciences (SPSS) version 26 (IBM Corp, Armonk, NY, USA). After normality adjustments using the Wilk-Shapiro test, quantitative variables were presented as mean ± standard deviation (SD) for normal distribution and median (Interquartile range: IQR) for non-normal distribution. Qualitative variables were presented as frequency and percentages. Comparison of the four groups was analyzed by Student's t-test and Chi-square test was used for qualitative variables. For outcomes assessment, cumulative incidence and Kaplan Meier curves were plotted for all four groups against the number of days in the hospital and mortality rate. Odds ratio (OR) and 95% confidence interval (CI) were analyzed for the severity of COVID-19 disease, mortality, and hospital stay. Correlation between blood group and lab parameters was presented as histogram distributed among the four groups. Logistic regression was done for predictors of severe COVID-19 disease for ABO blood groups. A p-value of less than 0.05 was considered significant.

## Results

A total of 1067 patients were admitted to Advanced Diagnostics and Liver Center from April 2020 through January 2021 who complied with our study criteria for analysis. The mean age of the patients in group 1, 2, 3, and 4 was 47.37 ± 20.21, 47.71 ± 18.45, 47.54 ± 18.84, and 48.87 ± 21.31 years, respectively. The majority of the patients were males 712 (66.7%) with an average body mass index (BMI) of 27.45 ± 3.53. Overall, one-quarter of the patients were diabetic 310 (29%), had hypertension 361 (33.8%), cardiovascular disease (CVD) 282 (26.4%), and 96 (8.9%) had chronic kidney disease (CKD). The majority of blood groups in descending order for this cohort were O (48.8%), B (27.6%), A (18.9%), and 4.5% were AB blood type.

The average median (IQR) hospital stay was 12 (6,25) days. Patients with AB blood group stayed a median (IQR) of 14 (5, 27) days while A blood group cohort stayed 13 (6,27) days. It was statistically non-significant between all four groups. There was overall 10.6% COVID-19-related mortality at our center, with 13.9% in blood group A as the majority of COVID-19 deaths. However, this did not reach statistical significance. Regarding severity of COVID-19 disease, there was a trend towards critical disease in blood group A (n = 83, 41.1%; OR 0.257 (95% CI: 0.167–1.136); p = 0.278) and O (n = 183, 35.1%; OR 0.438 (95% CI: 0.168–1.139); p = 0.090). However, the results were insignificant. Baseline characteristics, patient demographics, the severity of COVID-19, total hospital stay, and mortality rates are presented in **Table 1**. The overall incidence of the ABO blood type in the COVID-19 cohort and cumulative hazard of cause-specific death is demonstrated in **Fig 1** and the levels of acute phase reactants (IL-6, CRP, Procalcitonin, D-dimers) with ABO blood type is exhibited in **Fig 2**. IL-6 was

**Table 1. Patient demographics and baseline characteristics of COVID-19 patients with different blood groups.**

| Variable | Blood Group A (n = 202) | Blood Group B (n = 295) | Blood Group O (n = 521) | Blood Group AB (n = 49) | OR (95% CI) | P-value |
|---|---|---|---|---|---|---|
| **Age, years, mean ± SD** | 47.37 ± 20.21 | 47.71 ± 18.45 | 47.54 ± 18.84 | 48.87 ± 21.31 | - | 0.444 |
| **Gender, n (%)** | | | | | | 0.342 |
| Male | 138 (62.4%) | 239 (69.3%) | 283 (67.5%) | 52 (63.4%) | - | |
| Female | 83 (37.6%) | 106 (30.7%) | 136 (32.5%) | 30 (36.6%) | - | |
| **BMI, kg/m$^2$, mean ± SD** | 27.34 ± 3.48 | 27.72 ± 3.67 | 27.35 ± 3.76 | 27.48 ± 3.22 | - | 0.966 |
| **Comorbid conditions, n (%)** | | | | | | |
| DM | 67 (30.3%) | 97 (28.1%) | 119 (28.4%) | 27 (32.9%) | - | 0.744 |
| HTN | 80 (36.2%) | 106 (30.7%) | 146 (34.8%) | 29 (35.4%) | - | 0.391 |
| CVD | 61 (27.6%) | 102 (29.6%) | 95 (22.7%) | 24 (29.3%) | - | 0.407 |
| CKD | 20 (9%) | 33 (9.6%) | 38 (9.1%) | 5 (6.1%) | - | 0.985 |
| **COVID-19 severity, n (%)** | | | | | | |
| Mild | 51 (25.2%) | 82 (27.8%) | 156 (29.9%) | 30 (61.2%) | 6.27 (9.09–9.77) | **<0.001** |
| Moderate | 68 (33.7%) | 128 (43.4%) | 182 (34.9%) | 14 (28.6%) | 4.88 (19.98–21.55) | **0.036** |
| Critical | 83 (41.1%) | 86 (29.2%) | 183 (35.1%) | 5 (10.2%) | 11.34 (46.79–53.22) | **<0.001** |
| **Hospital stay, median (IQR)** | 13 (6,27) | 12 (6,25) | 12 (5,24) | 14 (5,27) | 3.78 (44.03–50.18) | 0.364 |
| **Mortality, n (%)** | 28 (13.9%) | 28 (9.5%) | 52 (10%) | 5 (10.2%) | 2.38 (30.28–34.71) | 0.412 |
| **Lab parameters, median (IQR)** | | | | | | |
| Hgb (g/dl) | 12.7 (10.9,13.8) | 12.3 (10.7,14.1) | 12.7 (11.2,14.8) | 12.7 (11.1,15) | - | 0.377 |
| WBC (mm$^3$) | 12000 (7000,15000) | 11000 (6000,14000) | 12000 (7000,15000) | 10500 (6000,15000) | - | 0.707 |
| IL-6 (pg/ml) | 23.6 (17.5,43.8) | 21.7 (17.5,43.8) | 19.7 (16.4,28.7) | 19.7 (16.7,32.4) | - | **0.001** |
| Procalcitonin (ng/ml) | 0.5 (0.3,0.65) | 0.54 (0.3,0.7) | 0.45 (0.23,0.67) | 0.5 (0.23,0.65) | - | 0.348 |
| D-dimers (mcg/ml) | 15 (8,32) | 15 (7,33) | 15 (5,28) | 21.5 (9,34) | - | 0.556 |
| CRP (mg/dl) | 17 (11,54) | 23 (9,60) | 17 (9,54) | 24 (9,49) | - | 0.146 |

Normally distributed variables expressed as mean ± SD, abnormally distributed variables expressed as median (IQR). Categorical variables presented as n (%). P < 0.05 as significant. Standard deviation (SD), interquartile range (IQR), body mass index (BMI), diabetes mellitus (DM), hypertension (HTN), cardiovascular disease (CVD), chronic kidney disease (CKD), hemoglobin (Hgb), white blood count (WBC), C-reactive protein (CRP), interquartile range (IQR), standard deviation (SD), odds ratio (OR), confidence interval (CI).

elevated in blood group A (median [IQR]: 23.6 [17.5,43.8]), Procalcitonin in blood group B (median [IQR]: 0.54 [0.3,0.7]), D-dimers and CRP in group AB (median [IQR]: 21.5 [9,34]; 24 [9,49], respectively).

Multivariate analysis (after adjusting age, gender, and comorbidities) did not show any blood group to be significantly associated with disease severity and mortality in this cohort (**Tables 2 and 3**). The cause-specific hazards ratio (HR) for survival function was 3.206 (p = 0.361) among all blood groups (**Fig 1**). More survival was seen with blood group A initially but with a hospital stay of more than approximately 30 days, the survival decreased exponentially in blood group A, while early mortality was observed with blood group AB (**Fig 1**).

## Discussion

Of all the human blood group systems, the most widely used in clinical practice is the ABO blood group and includes four blood types, including A, AB, B, and O. It is located on

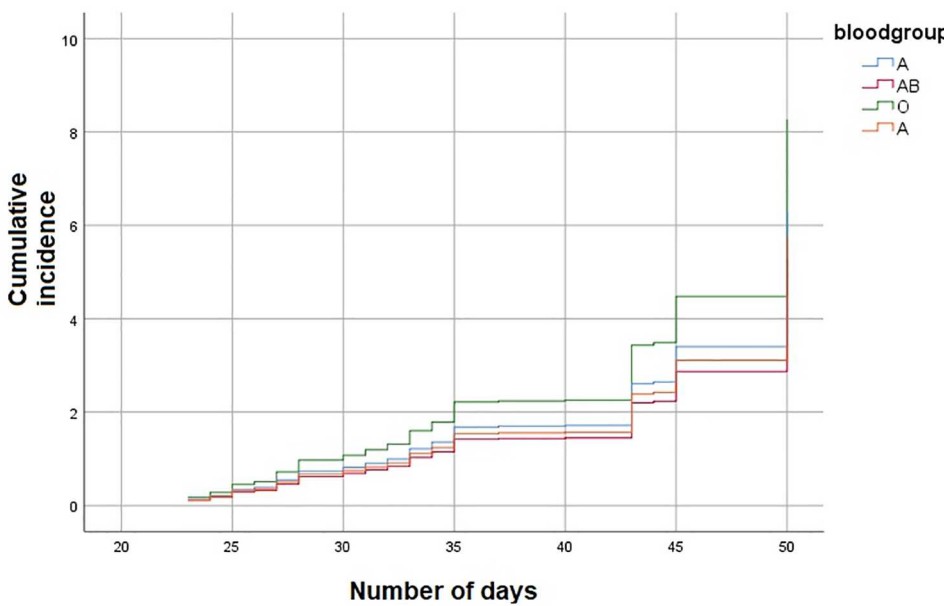

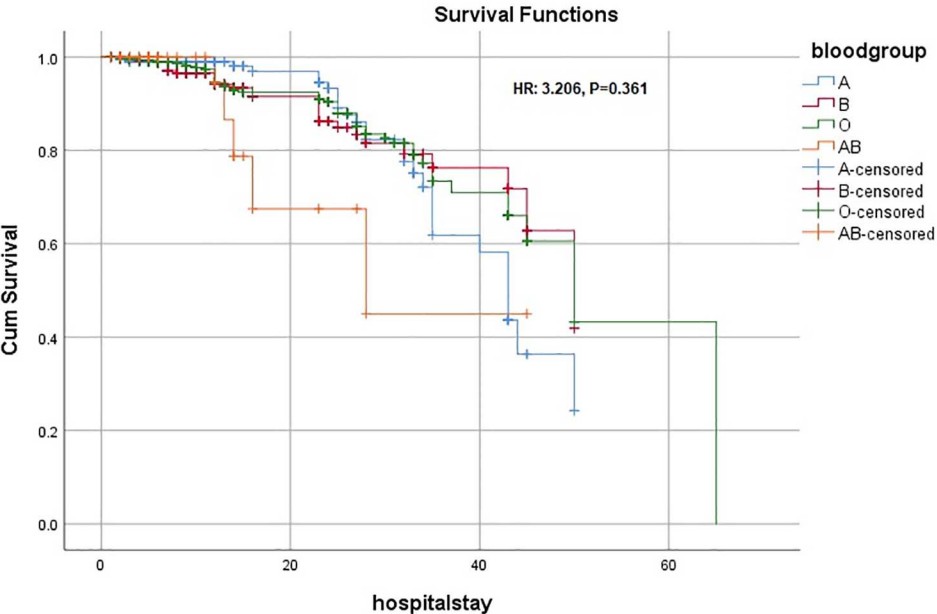

**Fig 1. Cumulative incidence and survival function Kaplan Meier curve of ABO blood type in COVID-19 cohort.**

chromosome 9 in a human DNA (9q34.2) and many studies have demonstrated a vital role of the ABO blood group in some infectious and non-infectious diseases. Histo-blood group antigens (HBGAs) are one of the main antigens expressed on human red blood cells, and differences in blood group antigens can alter host susceptibility to many infections. HBGAs are postulated to decrease the spread of infections through antibodies and ABO antibodies are a part of the innate immune system against many pathogens [10].

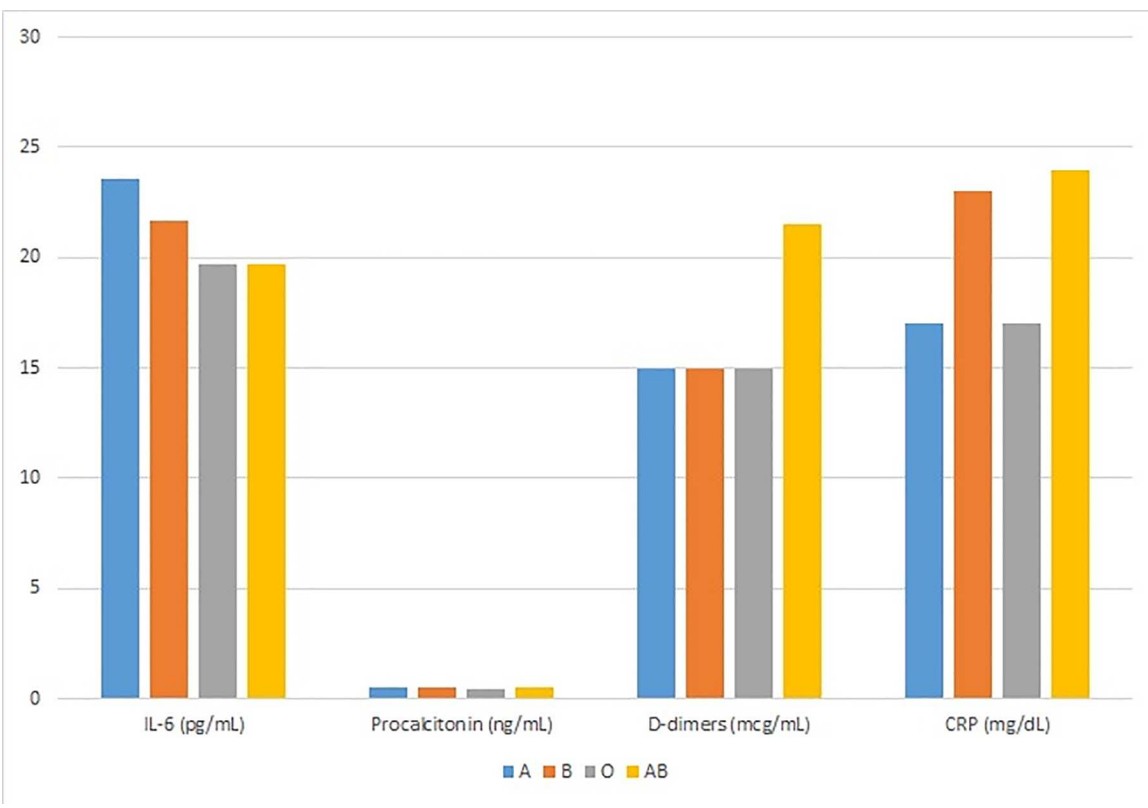

**Fig 2. Levels of acute phase reactants in correlation with ABO blood groups.**

In this study, we demonstrated a relationship of COVID-19 severity with ABO blood groups. Blood groups A and O had the majority of severe cases in this cohort of COVID-19 patients. By contrast, we observed mild disease to be prevalent in the O and AB blood groups. Acute phase reactants were not majorly deranged between the blood groups and hospital stay was increased with A and AB blood groups. Overall mortality was high at 10.5% and blood group A was associated with the highest COVID-19-associated mortality. The majority of blood group reported in this cohort was O (48.8%).

There are several studies demonstrating a relationship between ABO blood groups and COVID-19 disease. A study from China tested the association of ABO blood group with COVID-19 infection on 105 COVID-19 cases and 103 controls. Blood group A was prevalent in their population (42.8%) and it was statistically associated with increased infection risk of COVID-19 (OR: 1.33, 95% CI: 1.02–1.73, p = 0.04) in their female population [13]. Two recent studies from the subcontinent: one from Peshawar, Pakistan, and the other from Dhaka,

**Table 2. Multivariate analysis on ABO blood groups for associated mortality COVID-19 disease.**

| Blood group | B | SE | Wald ($t^2$) | aOR (95% CI) | P-value |
|---|---|---|---|---|---|
| A | -0.576 | 0.491 | 3.852 | 0.257 (0.167–1.136) | 0.278 |
| B | -0.680 | 0.489 | 1.935 | 0.507 (0.194–1.321) | 0.164 |
| O | -0.826 | 0.488 | 2.869 | 0.438 (0.168–1.139) | 0.090 |
| AB | -0.879 | 0.471 | 3.479 | 0.415 (0.165–1.046) | 0.062 |

Adjusted odds ratio (aOR), confidence interval (CI), beta (B), standard error (SE), chi square distributed with df = 1. P<0.05 as significant.

**Table 3. Multivariate analysis on ABO blood groups for severity of COVID-19 disease.**

| Blood group | B | SE | Wald ($t^2$) | aOR (95% CI) | P-value |
|---|---|---|---|---|---|
| A | -0.632 | 0.563 | 1.654 | 0.164 (0.132–0.736) | 0.165 |
| B | -0.532 | 0.163 | 1.732 | 0.705 (0.276–1.943) | 0.214 |
| O | -0.814 | 0.254 | 2.623 | 0.743 (0.143–1.642) | 0.079 |
| AB | -0.632 | 0.545 | 2.732 | 0.154 (0.125–0.953) | 0.165 |

Adjusted odds ratio (aOR), confidence interval (CI), beta (B), standard error (SE), chi square distributed with df = 1. P<0.05 as significant.

Bangladesh have demonstrated the susceptibility of COVID-19 with the ABO blood groups [15, 16]. The study from Pakistan had a sample size of 1935 patients, with blood group B as the prevalent blood type (35.9%) with an increased susceptibility for COVID-19 infection (OR: 1.195 (95% CI: 1.04–1.36), p = 0.009) while blood groups A and O did not have statistically significant association of positive RT-PCR for SARS-COV-2. The other study from Bangladesh included 381 patients with a prevalence of blood group A in the COVID-19 cohort (32.9%, p<0.001), and no significant differences were observed in the duration of symptoms among other blood types. Our study exhibits contrasting results when compared to these investigations. Blood group O was prevalent in our study cohort (48.8%) and severe disease was associated with blood group A and O (41.1%, 35.1%, respectively). Hospital stay was more with blood type A and AB and higher mortality was associated with blood type A as well (13.9%).

In our previous study and a preprint, we demonstrated altered lipid profiles and thyroid function tests in association with COVID-19 disease and its severity along with acute phase reactants [17, 18]. Acute phase reactants were severely deranged in both study cohorts and there was a positive correlation with the higher classification of severity of COVID-19. A similar conclusion was given in a systematic review of 34 articles, showing the derangement of laboratory parameters with increasing severity of COVID-19 [19]. In this investigation, IL-6 and D-dimers were elevated in blood group A, Procalcitonin in blood group B, and CRP in blood group O. Similarly, a study from Canada reported no difference in acute phase reactants between blood groups A, AB, O, or B [20]. This phenomenon is still unclear with heterogeneous results in different populations and needs further work to understand the underlying mechanisms.

There were several limitations to this study. First, it was a single-center study and the retrospective nature limits the control of confounding factors. Second, due to the limited sample size of COVID-19 in the early stages, the sample size included in this study was not very large. Third, as it represented the population of one province, so a regional selection bias needs to be considered. Third, other comorbidities might influence the severity of the disease. Fourth, outcomes of the patients' treatment were not examined in this study.

## Conclusion

In conclusion, this investigation did not show significant association of blood groups with severity and of COVID-19 disease and COVID-19-associated mortality. Alteration of the acute phase reactants was not positively associated with any specific blood type.

## Supporting information

**S1 File.**
(SAV)

## Author Contributions

**Conceptualization:** Uzma Ishaq, Jahanzeb Malik.

**Data curation:** Uzma Ishaq, Jahanzeb Malik, Asad Mehmood, Azhar Qureshi, Talha Laique, Syed Muhammad Jawad Zaidi, Abdul Sattar Rana.

**Formal analysis:** Jahanzeb Malik, Syed Muhammad Jawad Zaidi, Abdul Sattar Rana.

**Methodology:** Asmara Malik, Jahanzeb Malik, Abdul Sattar Rana.

**Project administration:** Uzma Ishaq, Jahanzeb Malik.

**Writing – original draft:** Uzma Ishaq, Asmara Malik, Jahanzeb Malik, Asad Mehmood, Azhar Qureshi, Talha Laique, Syed Muhammad Jawad Zaidi, Muhammad Javaid.

**Writing – review & editing:** Asmara Malik, Jahanzeb Malik, Syed Muhammad Jawad Zaidi, Muhammad Javaid.

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
