## [Decision Letter · Decision Letter 0]

21 Jun 2021

PONE-D-21-18070

Association of ABO blood group with COVID-19 severity, acute phase reactants and mortality

PLOS ONE

Dear Dr. Malik,

Thank you for submitting your manuscript to PLOS ONE. After careful consideration, we feel that it has merit but does not fully meet PLOS ONE’s publication criteria as it currently stands. Therefore, we invite you to submit a revised version of the manuscript that addresses the points raised during the review process.

Please address the issues and revise accordingly if able.

We look forward to receiving your revised manuscript.

Kind regards,

Academic Editor

PLOS ONE

Journal Requirements:

2. Please include the tables within the main manuscript.

Reviewers' comments:

Reviewer's Responses to Questions

**Comments to the Author**

1. Is the manuscript technically sound, and do the data support the conclusions?

Reviewer #1: No

Reviewer #2: No

2. Has the statistical analysis been performed appropriately and rigorously? 

Reviewer #1: No

Reviewer #2: No

3. Have the authors made all data underlying the findings in their manuscript fully available?

Reviewer #1: Yes

Reviewer #2: Yes

4. Is the manuscript presented in an intelligible fashion and written in standard English?

Reviewer #1: No

Reviewer #2: No

5. Review Comments to the Author

Reviewer #1: This study entitled “Association of ABO blood group with COVID-19 severity, acute phase reactants and Mortality” retrospectively analyzed the association between blood ABO groups and disease severity/mortality among the COVID-19 patients.

Some comments and questions were raised for the authors.

1. Many confounding factors would affect the clinical outcome (severity and mortality), such as comorbidities and medical treatment. Therefore, to minimize confounding bias, a multivariate analysis which includes variables of blood groups, comorbidities, and treatment should be performed to determine the risk factors associated with disease severity or mortality. If a certain blood group remains a risk factor for severe disease or mortality after a multivariate analysis, it would be more convincing to state that certain blood group is associated with severe disease and higher mortality.

2. The presentation and interpretation of some statistical results were not precise or not consistent throughout the manuscript. For example,

(1) Abstract, conclusions, “…hospital stay, severity of disease, and mortality were associated with blood group A.” -> However, hospital stay was not different between all four blood groups as mentioned in the Result section “Patients with AB blood group stayed a median (IQR) of 14 (5, 27) days while A blood group cohort stayed 13 (6,27) days. It was statistically non-significant between all four groups.” In addition, the mortality rates among 4 blood groups were not statistically different according to Table 1.

(2) Results: “Regarding severity of COVID-19 disease, there was a trend towards critical disease in blood groups A and O (n=83, 41.1%; n=183, 35.1%; OR, 11.34 (95% CI, 46.79-53.22); p<0.001).” Why was OR ratio here beyond the range of 95% CI? In addition, to determine the effect of blood group A and O on disease severity, blood group A and O should have individual OR.

(3) Discussion, “By contrast, we observed mild disease to be prevalent in the B and AB blood groups.“ -> The statement is not consistent with the data in Table 1.

(4) Discussion, “Acute phase reactants were majorly deranged in blood group O as compared to other groups and hospital stay was associated more with A and AB blood groups”. -> However, according to Figure 2, blood group O appeared to have the highest levels of acute phase proteins, whereas blood group O had the lowest levels of acute phase proteins in Table 1. Please check these data again.

(5) Conclusions, “In conclusion, we demonstrate that male patients with blood group A and O are associated with an increased risk of severe COVID-19 disease after gender stratification while group A is associated with increased hospital stay and mortality.”-> According to the context, the effect of gender was not examined in this study. Furthermore, the association of group A with increased hospital stay and mortality was not evident according to Table 1.

(6) What did “the number of days” mean in the figure 1 (cumulative incidence vs numbers of days)? What’s the unit for incidence here?

Overall, it would be helpful if the statistical analysis of this study is carefully reviewed again by a statistics expert.

Minor comments:

1. Method, “The severity of COVID-19 was classified into, mild, moderate, and critical according to the Centers for Disease Control (CDC) guidelines.” -> Please provide reference here.

2. Please define the headings of Table 3 (B, SE, Wald).

Reviewer #2: Several comments,

I. In general, the English grammar should be checked before submission

II. In the introduction, the authors should clearly describe the current status of studies about the association between ABO blood types and COVID-19. Both pros and cons options about this issue, especially severity and mortality, should also be cited in this section. Moreover, this section also lacks flow and transition sentences between paragraphs.

III. Materials and Methods

A. The definitions of COVID-19 severity are incorrect.

B. How to define COVID-19 related mortality?

IV. Results

A. How about the distribution of comorbidities (other than those described by authors) between different blood types?

B. The therapeutic strategies between patients with different blood types should be pointed out.

V. Discussion

A. The second paragraph only describes the results of this study. No comparison was made with prior studies.

B. In the third paragraph, the authors described different prevalent results of blood types among various studies, including themselves. More discussion should be made about such issues.

VI. Tables and figures

A. Table 2 is not clear

B. The resolution of figures is poor

6. PLOS authors have the option to publish the peer review history of their article (what does this mean?). If published, this will include your full peer review and any attached files.

Reviewer #1: No

Reviewer #2: No

---

## [Author Response · Author response to Decision Letter 0]

16 Jul 2021

Dear editors and reviewers

Thank you for taking out time to comment on this manuscript. Your comments were valuable in making this manuscript better.

Point to point response 

Reviewer 1

Issue 1: Many confounding factors would affect the clinical outcome (severity and mortality), such as comorbidities and medical treatment. Therefore, to minimize confounding bias, a multivariate analysis which includes variables of blood groups, comorbidities, and treatment should be performed to determine the risk factors associated with disease severity or mortality. If a certain blood group remains a risk factor for severe disease or mortality after a multivariate analysis, it would be more convincing to state that certain blood group is associated with severe disease and higher mortality.

Response: We re-evaluated the data and did multivariate analysis with adjusting comorbid conditions and age/gender

Issue 2: Abstract, conclusions, “…hospital stay, severity of disease, and mortality were associated with blood group A.” -> However, hospital stay was not different between all four blood groups as mentioned in the Result section “Patients with AB blood group stayed a median (IQR) of 14 (5, 27) days while A blood group cohort stayed 13 (6,27) days. It was statistically non-significant between all four groups.” In addition, the mortality rates among 4 blood groups were not statistically different according to Table 1.

Response: we corrected the abstract as recommended.plz see

Issue 3: Results: “Regarding severity of COVID-19 disease, there was a trend towards critical disease in blood groups A and O (n=83, 41.1%; n=183, 35.1%; OR, 11.34 (95% CI, 46.79-53.22); p<0.001).” Why was OR ratio here beyond the range of 95% CI? In addition, to determine the effect of blood group A and O on disease severity, blood group A and O should have individual OR.

Response: OR corrected and added for both blood groups and mistakes rectified. Plz see

Issue 4: Discussion, “By contrast, we observed mild disease to be prevalent in the B and AB blood groups.“ -> The statement is not consistent with the data in Table 1.

Response: Very avid observation dear sir, we corrected the mistakes. Thank you for commenting. Plz see

Issue 5: Discussion, “Acute phase reactants were majorly deranged in blood group O as compared to other groups and hospital stay was associated more with A and AB blood groups”. -> However, according to Figure 2, blood group O appeared to have the highest levels of acute phase proteins, whereas blood group O had the lowest levels of acute phase proteins in Table 1. Please check these data again.

Response: we rechecked the data and new table is made with mean acute phae reactant values by our statistician. Plz see updated figure

Issue 6: Conclusions, “In conclusion, we demonstrate that male patients with blood group A and O are associated with an increased risk of severe COVID-19 disease after gender stratification while group A is associated with increased hospital stay and mortality.”-> According to the context, the effect of gender was not examined in this study. Furthermore, the association of group A with increased hospital stay and mortality was not evident according to Table 1.

Response: Thankyou dear sir for the observation. You are correct. We rectified the mistake

Minor comments:

1. Method, “The severity of COVID-19 was classified into, mild, moderate, and critical according to the Centers for Disease Control (CDC) guidelines.” -> Please provide reference here.

2. Please define the headings of Table 3 (B, SE, Wald).

Response: updated classification and reference added and table 2 headings dfined in table legend

Reviewer 2

Issue 1: I. In general, the English grammar should be checked before submission

Response: English syntax rechecked 

Issue 2: In the introduction, the authors should clearly describe the current status of studies about the association between ABO blood types and COVID-19. Both pros and cons options about this issue, especially severity and mortality, should also be cited in this section. Moreover, this section also lacks flow and transition sentences between paragraphs.

Response: updated investigations added to intro for a better clarity and syntax imporved

Issue 3: Materials and Methods

A. The definitions of COVID-19 severity are incorrect.

B. How to define COVID-19 related mortality?

Response: definition update for severity classification of covid 19, and any death in which no other cause was apparent other than covid 19 was considered as covid associated mortality

Issue 4: IV. Results

A. How about the distribution of comorbidities (other than those described by authors) between different blood types?

B. The therapeutic strategies between patients with different blood types should be pointed out.

Response: dear sit I regret to inform that no other comorbidities were collected in our study and no therapeutic strategy was noted. That data is not available to us unfortunately.

Issue 5: Discussion

A. The second paragraph only describes the results of this study. No comparison was made with prior studies.

B. In the third paragraph, the authors described different prevalent results of blood types among various studies, including themselves. More discussion should be made about such issues.

Response: Discussion altered a little for more clarity

Issue 6; Tables and figures

A. Table 2 is not clear

B. The resolution of figures is poor

Figures cleared and table legends added

Editorial comments

Please include the tables within the main manuscript.

Included in the main text

Please note that in order to use the direct billing option the corresponding author must be affiliated with the chosen institute. Please either amend your manuscript to change the affiliation or corresponding author, or email us at plosone@plos.org with a request to remove this option.

Affiliation amended

Your ethics statement should only appear in the Methods section of your manuscript. If your ethics statement is written in any section besides the Methods, please move it to the Methods section and delete it from any other section. Please ensure that your ethics statement is included in your manuscript, as the ethics statement entered into the online submission form will not be published alongside your manuscript.

Ethics statement added to methods

---

## [Decision Letter · Decision Letter 1]

28 Jul 2021

PONE-D-21-18070R1

Association of ABO blood group with COVID-19 severity, acute phase reactants and mortality

PLOS ONE

Dear Dr. Malik,

Thank you for submitting your manuscript to PLOS ONE. After careful consideration, we feel that it has merit but does not fully meet PLOS ONE’s publication criteria as it currently stands. Therefore, we invite you to submit a revised version of the manuscript that addresses the points raised during the review process.

Please address the issues and revise accordingly.

We look forward to receiving your revised manuscript.

Kind regards,

Academic Editor

PLOS ONE

Reviewers' comments:

Reviewer's Responses to Questions

**Comments to the Author**

1. If the authors have adequately addressed your comments raised in a previous round of review and you feel that this manuscript is now acceptable for publication, you may indicate that here to bypass the “Comments to the Author” section, enter your conflict of interest statement in the “Confidential to Editor” section, and submit your "Accept" recommendation.

Reviewer #1: (No Response)

Reviewer #3: All comments have been addressed

2. Is the manuscript technically sound, and do the data support the conclusions?

Reviewer #1: No

Reviewer #3: No

3. Has the statistical analysis been performed appropriately and rigorously? 

Reviewer #1: No

Reviewer #3: Yes

4. Have the authors made all data underlying the findings in their manuscript fully available?

Reviewer #1: Yes

Reviewer #3: Yes

5. Is the manuscript presented in an intelligible fashion and written in standard English?

Reviewer #1: No

Reviewer #3: Yes

6. Review Comments to the Author

Reviewer #1: In general, the presentation and interpretation of statistical results was still not precise in this revised manuscript.

Some comments and questions are listed below:

- Abstract: “Acute phase protein reactants” was included in the manuscript title; however, nothing was mentioned in the abstract.

- Abstract (Method): Information provided in the method was very limited.

- Abstract (Results): “Patients with AB blood group stayed a median (IQR) of 14 (5, 27) days while A blood group cohort stayed 13 (6,27) days and overall…” It is difficult to understand why the authors mentioned days of hospital stay for AB blood group and A blood group here, as the days of hospital stay were not significantly different between 4 groups. Why was AB blood group mentioned here in particular?

- Abstract (Results): “…overall 10.6% COVID-19-related mortality was observed at our center, with 13.9% in blood group A as the majority of COVID-19 deaths.” This part could be written as a separate sentence. In addition, although patients of blood group A had a higher proportion of mortality, patients with blood group O contributed to the majority of deaths.

- Abstract & Results: “Regarding severity of COVID-19 disease, there was a trend towards critical disease in blood group A (n=83, 41.1%; OR 0.257 (95% CI:0.167-1.136); p<0.001) and O (n=183, 35.1%; OR 0.438 (95% CI: 0.168-1.139); p<0.001).” The OR (95% CI) presented in this sentence were OR (95% CI) for “mortality” shown in Table 2 (multivariate analysis), but the proportions and p values presented in this sentence were the proportions and p values for critical diseases shown in Table 1 (univariate analysis). The presentation and interpretation of statistical results is confusing.

Result

- Please briefly describe the results of acute phase protein reactants in the text.

- “Multivariate analysis (after adjusting age, gender, and comorbidities) did not show any blood group in this cohort (Table 2).” This sentence was not clear.

- As the study aimed to exam the association between the blood group and disease severity, multivariate analysis should also be performed to identify risk factors associated with severity (as done in Table 2 which examined the association between the blood group and mortality by multivariate analysis.)

Discussion

- “, blood group O was linked to severe derangement of acute-phase reactants.” Data from Table 1 and Figure 2did not support this statement.

Conclusion

- “Alteration of the acute phase reactants is positively associated with blood type A.” Data from Table 1 and Figure 2 did not support this statement, and this statement was not consistent with “blood group O was linked to severe derangement of acute-phase reactants” mentioned in the Discussion section.

Figure

The resolution of Figure 2 was not enough.

Reviewer #3: Reviewer 2 asked

A. How about the distribution of comorbidities (other than those described by authors)

between different blood types?

B. The therapeutic strategies between patients with different blood types should be

pointed out.

Response: dear sit I regret to inform that no other comorbidities were collected in our

study and no therapeutic strategy was noted. That data is not available to us

unfortunately.

These are major confounding factors and should be assessed.

Other confounding factor is the time from the onset of the disease to hospital admission. It should be included to the analysis.

Please provide rationale support to assuring adequate sample power.

The authors stated that p"articipants had blood group O as the prevalent blood type." However this percentage should be compared with the distribution of blood groups in the country.

7. PLOS authors have the option to publish the peer review history of their article (what does this mean?). If published, this will include your full peer review and any attached files.

Reviewer #1: No

Reviewer #3: No

---

## [Author Response · Author response to Decision Letter 1]

27 Aug 2021

Dear editors and reviewers

Thank you for taking out time to comment on this manuscript. Your comments were valuable in making this manuscript better.

Point to point response 

Abstract: “Acute phase protein reactants” was included in the manuscript title; however, nothing was mentioned in the abstract.

Response: abstract corrected and acute phase protein results added

Abstract (Method): Information provided in the method was very limited.

Response: method info increased as advised

Abstract (Results): “Patients with AB blood group stayed a median (IQR) of 14 (5, 27) days while A blood group cohort stayed 13 (6,27) days and overall…” It is difficult to understand why the authors mentioned days of hospital stay for AB blood group and A blood group here, as the days of hospital stay were not significantly different between 4 groups. Why was AB blood group mentioned here in particular?

Response: results amended. Please see

Abstract (Results): “…overall 10.6% COVID-19-related mortality was observed at our center, with 13.9% in blood group A as the majority of COVID-19 deaths.” This part could be written as a separate sentence. In addition, although patients of blood group A had a higher proportion of mortality, patients with blood group O contributed to the majority of deaths.

Response: statement amended

Abstract & Results: “Regarding severity of COVID-19 disease, there was a trend towards critical disease in blood group A (n=83, 41.1%; OR 0.257 (95% CI:0.167-1.136); p<0.001) and O (n=183, 35.1%; OR 0.438 (95% CI: 0.168-1.139); p<0.001).” The OR (95% CI) presented in this sentence were OR (95% CI) for “mortality” shown in Table 2 (multivariate analysis), but the proportions and p values presented in this sentence were the proportions and p values for critical diseases shown in Table 1 (univariate analysis). The presentation and interpretation of statistical results is confusing.

Response: p values corrected

“Multivariate analysis (after adjusting age, gender, and comorbidities) did not show any blood group in this cohort (Table 2).” This sentence was not clear.

Response: Sentence corrected and Multivariate of disease severity added in table 3

blood group O was linked to severe derangement of acute-phase reactants.” Data from Table 1 and Figure 2did not support this statement.

Response: all statements regarding acute phase reactants corrected please see

Alteration of the acute phase reactants is positively associated with blood type A.” Data from Table 1 and Figure 2 did not support this statement, and this statement was not consistent with “blood group O was linked to severe derangement of acute-phase reactants” mentioned in the Discussion section

Response: all statements regarding acute phase reactants corrected please see

Regards

---

## [Decision Letter · Decision Letter 2]

6 Sep 2021

PONE-D-21-18070R2Association of ABO blood group with COVID-19 severity, acute phase reactants and mortalityPLOS ONE

Dear Dr. Malik,

Thank you for submitting your manuscript to PLOS ONE. After careful consideration, we feel that it has merit but does not fully meet PLOS ONE’s publication criteria as it currently stands. Therefore, we invite you to submit a revised version of the manuscript that addresses the points raised during the review process.

Please address the concerns of the reviewer with unfavorable opinions and revise the manuscript.  If not amendable, maybe put into the Limitations.

We look forward to receiving your revised manuscript.

Kind regards,

Academic Editor

PLOS ONE

Reviewers' comments:

Reviewer's Responses to Questions

**Comments to the Author**

1. If the authors have adequately addressed your comments raised in a previous round of review and you feel that this manuscript is now acceptable for publication, you may indicate that here to bypass the “Comments to the Author” section, enter your conflict of interest statement in the “Confidential to Editor” section, and submit your "Accept" recommendation.

Reviewer #1: (No Response)

Reviewer #4: All comments have been addressed

Reviewer #5: All comments have been addressed

2. Is the manuscript technically sound, and do the data support the conclusions?

Reviewer #1: No

Reviewer #4: Yes

Reviewer #5: Yes

3. Has the statistical analysis been performed appropriately and rigorously? 

Reviewer #1: I Don't Know

Reviewer #4: Yes

Reviewer #5: Yes

4. Have the authors made all data underlying the findings in their manuscript fully available?

Reviewer #1: Yes

Reviewer #4: Yes

Reviewer #5: Yes

5. Is the manuscript presented in an intelligible fashion and written in standard English?

Reviewer #1: No

Reviewer #4: Yes

Reviewer #5: Yes

6. Review Comments to the Author

Reviewer #1: All the earlier comments have been addressed by the authors. However, the interpretation and presentation of some results still appeared inaccurate. In many instances, some minor differences existed between groups without statistical differences, but they were presented as meaningful findings. Some expressions are also confusing, for example, “the frequency of acute phase reactants with ABO blood type” in text and “Frequency distribution of acute phase reactants in correlation with ABO blood group” in Figure 2. In addition, the data of acute phase reactants in table 1 looked quite different with those in Figure 2.

Reviewer #4: (No Response)

Reviewer #5: (No Response)

7. PLOS authors have the option to publish the peer review history of their article (what does this mean?). If published, this will include your full peer review and any attached files.

Reviewer #1: No

Reviewer #4: **Yes: **Parisa Sabbagh

Reviewer #5: **Yes: **Hamed Kalani

---

## [Author Response · Author response to Decision Letter 2]

27 Sep 2021

Dear editors and reviewers

Thank you for taking out time to comment on this manuscript. Your comments were valuable in making this manuscript better.

Point to point response 

Question: In many instances, some minor differences existed between groups without statistical differences, but they were presented as meaningful findings.

Response: all minor differences corrected and meaningless findings excluded from the manuscript

Question: Some expressions are also confusing, for example, “the frequency of acute phase reactants with ABO blood type” in text and “Frequency distribution of acute phase reactants in correlation with ABO blood group” in Figure 2.

Response: all confusing statements corrected

Question: In addition, the data of acute phase reactants in table 1 looked quite different with those in Figure 2.

Response: figure and table matched. Please see

Regards

---

## [Decision Letter · Decision Letter 3]

22 Nov 2021

PONE-D-21-18070R3Association of ABO blood group with COVID-19 severity, acute phase reactants and mortalityPLOS ONE

Dear Dr. Malik,

Thank you for submitting your manuscript to PLOS ONE. After careful consideration, we feel that it has merit but does not fully meet PLOS ONE’s publication criteria as it currently stands. Therefore, we invite you to submit a revised version of the manuscript that addresses the points raised during the review process. Please revise.

We look forward to receiving your revised manuscript.

Kind regards,

Academic Editor

PLOS ONE

Journal Requirements:

Reviewers' comments:

Reviewer's Responses to Questions

**Comments to the Author**

1. If the authors have adequately addressed your comments raised in a previous round of review and you feel that this manuscript is now acceptable for publication, you may indicate that here to bypass the “Comments to the Author” section, enter your conflict of interest statement in the “Confidential to Editor” section, and submit your "Accept" recommendation.

Reviewer #4: (No Response)

Reviewer #5: All comments have been addressed

2. Is the manuscript technically sound, and do the data support the conclusions?

Reviewer #4: Yes

Reviewer #5: Yes

3. Has the statistical analysis been performed appropriately and rigorously? 

Reviewer #4: Yes

Reviewer #5: Yes

4. Have the authors made all data underlying the findings in their manuscript fully available?

Reviewer #4: Yes

Reviewer #5: Yes

5. Is the manuscript presented in an intelligible fashion and written in standard English?

Reviewer #4: Yes

Reviewer #5: Yes

6. Review Comments to the Author

Reviewer #4: Its a good and useful article. And also it has understandable and fluent text. Subject is updated and its one of the most important problem nowadays.

Reviewer #5: 1- Move sentence “Alteration of the acute phase reactants was not positively associated with any specific blood type” from section of Conclusion in Abstract to section of Results.

2- It should be noted in the discussion that the outcome of patients' treatment was not examined in this study and this is one of the limitations of this study.

3- Check for spelling and typographical errors throughout the text.

7. PLOS authors have the option to publish the peer review history of their article (what does this mean?). If published, this will include your full peer review and any attached files.

Reviewer #4: **Yes: **Parisa Sabbagh

Reviewer #5: No

---

## [Author Response · Author response to Decision Letter 3]

23 Nov 2021

Dear editors and reviewers

Thank you for taking out time to comment on this manuscript. Your comments were valuable in making this manuscript better.

Point to point response 

Question: Move sentence “Alteration of the acute phase reactants was not positively associated with any specific blood type” from section of Conclusion in Abstract to section of Results.

Response: Sentence moved to results

Question: It should be noted in the discussion that the outcome of patients' treatment was not examined in this study and this is one of the limitations of this study.

Response: Sentence added to limitations at the end of discussion

Question: Check for spelling and typographical errors throughout the text.

Response: Errors fixed

Regards

---

## [Decision Letter · Decision Letter 4]

2 Dec 2021

Association of ABO blood group with COVID-19 severity, acute phase reactants and mortality

PONE-D-21-18070R4

Dear Dr. Malik,

We’re pleased to inform you that your manuscript has been judged scientifically suitable for publication and will be formally accepted for publication once it meets all outstanding technical requirements.

Kind regards,

Academic Editor

PLOS ONE

Additional Editor Comments (optional):

Reviewers' comments:

Reviewer's Responses to Questions

**Comments to the Author**

1. If the authors have adequately addressed your comments raised in a previous round of review and you feel that this manuscript is now acceptable for publication, you may indicate that here to bypass the “Comments to the Author” section, enter your conflict of interest statement in the “Confidential to Editor” section, and submit your "Accept" recommendation.

Reviewer #4: All comments have been addressed

Reviewer #5: All comments have been addressed

2. Is the manuscript technically sound, and do the data support the conclusions?

Reviewer #4: Yes

Reviewer #5: Yes

3. Has the statistical analysis been performed appropriately and rigorously? 

Reviewer #4: Yes

Reviewer #5: Yes

4. Have the authors made all data underlying the findings in their manuscript fully available?

Reviewer #4: Yes

Reviewer #5: Yes

5. Is the manuscript presented in an intelligible fashion and written in standard English?

Reviewer #4: Yes

Reviewer #5: Yes

6. Review Comments to the Author

Reviewer #4: (No Response)

Reviewer #5: (No Response)

7. PLOS authors have the option to publish the peer review history of their article (what does this mean?). If published, this will include your full peer review and any attached files.

Reviewer #4: **Yes: **Parisa Sabbagh

Reviewer #5: No

---

## [Editor Report · Acceptance letter]

3 Dec 2021

PONE-D-21-18070R4 

Association of ABO blood group with COVID-19 severity, acute phase reactants and mortality 

Dear Dr. Malik:

I'm pleased to inform you that your manuscript has been deemed suitable for publication in PLOS ONE. Congratulations! Your manuscript is now with our production department. 

Kind regards, 

on behalf of

Dr. Robert Jeenchen Chen 

Academic Editor

PLOS ONE